# WATCHMAKER FUNCTIONS AND META SPECIFICATION OF OPEN-ENDED LEARNING SYSTEMS

## ABSTRACT

Open-ended learning systems aim to foster the continuous evolution of increasingly capable agents through the dynamic generation of novel challenges. The efficacy of these systems is fundamentally influenced by two critical factors: the design of the underlying system, which delineates the space of possibilities, and the open-ended algorithms that drive ongoing progress within this space. Current approaches to system design rely on explicit specification, where state spaces and evolution functions are fully defined at design time, often leading to prohibitive design complexity as systems scale. To address this challenge, we propose an alternative design principle termed *meta specification*. This approach defines systems implicitly through constraints, utilizing *watchmaker functions*—generalized stochastic evolution functions—coupled with verification routines to perform system evolution. Meta specification principles have the potential to significantly expand the space of possibilities while reducing design complexity, thereby enhancing the potential for open-ended learning. We demonstrate the viability of this principle through an illustrative implementation that co-evolves robot morphologies and robotic tasks, showcasing its capacity for emergent novelty and highlighting the shift in focus towards verification in system design.

## 1 INTRODUCTION

Recent advances in machine learning (ML), particularly in foundation models, have dramatically expanded the capabilities of autonomous agents across various domains. However, a key goal in ML research remains elusive: creating agents capable of continuous self-improvement. *Open-ended learning systems* (OELS) have emerged as a promising frontier to address this challenge (Soros et al., 2017; Clune, 2019). By dynamically generating novel challenges, OELS drive the adaptation of learning agents, creating environments that promote continuous exploration and skill acquisition. This approach enables agents to autonomously adapt to unforeseen scenarios and progressively enhance their capabilities beyond predefined limits (Jiang et al., 2023; Hughes et al., 2024).

The efficacy of an OELS in driving continuous progress and expanding agent capabilities hinges on two critical factors. First, the design of the underlying system plays a major role in delineating the *space of possibilities*, which we will formally introduce in Sec. 2. The system design effectively defines the boundaries within which learning and evolution can occur. Complementing this are open-ended algorithm that guide the system's evolution to produce continuous progress and novelty (Brant & Stanley, 2017; Hintze, 2019; Dennis et al., 2020; Zhang et al., 2024). The interplay between these factors underscores a crucial principle: while open-ended algorithms can enhance and sustain open-ended learning, the system's design ultimately imposes the *upper limit* on this potential. As a result, carefully designing and scaling the system to expand the space of possibilities is of critical importance for maximizing the potential of OELS (Team et al., 2021; Bauer et al., 2023).

Current approaches to system design predominantly rely on **explicit specification**, where the system's state spaces and evolution functions are fully defined at *design time* and kept fixed during its run time. Scaling the system under this paradigm typically involves introducing additional degrees of freedom into its design. This principle has yielded notable successes, as exemplified by Bauer et al. (2023), which demonstrated emergent capabilities and behaviors in a vast system encompassing 25 billion unique tasks. However, the design complexity associated with this approach increases dramatically with scale. As systems grow in scale, the explicit specification approach may become

prohibitively complex or intractable before reaching the level of complexity required to realize the full potential of OELS.

This work addresses the challenge of system design in OELS, specifically exploring methods to expand the space of possibilities while minimizing design complexity. We propose an alternative design principle termed **meta specification**. In contrast to explicit specification, which fully defines state spaces and evolution functions, meta specification defines a system *implicitly* through constraints placed on a generalized representation space. These constraints reflect both the fundamental requirements of the implementation platform (e.g. computing environment) and additional criteria set by the system designer. Implementing a system through meta specification necessitates three components: (1) a generalized representation for learning agents and tasks, (2) a mechanism to perform evolution over these representations, and (3) routines to verify constraint satisfaction. Among these, the generalized evolution mechanism presents the most significant challenge. To address this, we formally introduce the concept of **watchmaker functions**—classes of stochastic functions capable of performing meaningful transformations over generalized representation spaces—and establish the necessary conditions for these functions. Notably, we observe an intriguing connection between foundation models, particularly *Large Language Models* (LLMs), and the capabilities required of watchmaker functions, suggesting their potential as candidates for this role.

**Key considerations.** Implementing a system through meta specification involves designing its constituent components. While this principle may not be universally applicable to all open-ended learning objectives, it offers the potential to significantly expand the space of possibilities. Additionally, it shifts the focus of design complexity towards developing robust verification routines, potentially simplifying other aspects of system design. **Illustrative demonstration.** To assess the viability of this design principle, we present an illustrative implementation that co-evolves robot morphologies and robotic tasks using an LLM-based watchmaker function. This demonstration showcases emergent novelty in evolved robots and tasks that were not explicitly preprogrammed, and highlights the shift in emphasis towards verification in system design. Through this example, we illustrate the existence of key capabilities that could enable the extension of this principle to larger-scale implementations of OELS. Our core contributions are as follows:

1. We present a formal **unified framework** for conceptualizing OELS, providing a common language and structure for describing and comparing diverse OELS implementations.
2. We introduce a **novel design approach** for OELS systems based on meta specification principles. This approach leverages generalized representation spaces and watchmaker functions as evolution functions, integrating verification routines to implicitly define the system.
3. We provide an **illustrative demonstration** of the viability and potential of this design approach, presenting key ingredients that would enable its adoption in large-scale implementations.

## 2 UNIFIED FRAMEWORK FOR OELS

We begin by presenting a unified framework for *open-ended learning systems* (OELS), adopting a system-level perspective that distinguishes between two key components: the underlying **(1) dynamical system** (Sec. 2.1) and the **(2) control mechanism** (Sec. 2.2). Intuitively, the dynamical system is a coupled system of agents and tasks evolving over time. The control mechanism guides this system's evolution to be open-ended in nature, by monitoring the outputs of the system and performing control actions to guide the system towards continuous progress, as evaluated by some metric. To aid in exposition and contextualization, we will use POET (Wang et al., 2019) as a running example, and provide more comprehensive analysis of representative OELS using our framework in App. A. We provide a visual overview of our unified framework in Fig. 2.

### 2.1 DYNAMICAL SYSTEM COMPONENT

The underlying system in an OELS includes agents and tasks that evolve together. We formally define this system as a coupled dynamical system composed of agent and task subsystems, $\mathcal{S} = \langle \mathcal{S}_A, \mathcal{S}_T \rangle$, where $\mathcal{S}_A$ and $\mathcal{S}_T$ represent the **agent** and **task** subsystems, respectively.

1. **Agents.** Agents are the learning entities that evolve based on interactions with tasks. The subsystem is defined as the tuple $\mathcal{S}_A = \langle \mathcal{A}, \Phi_{\mathcal{A}} \rangle$. Here, $\mathcal{A}$ is the *state space* containing all possible agents (e.g. space of neural networks), where $a \in \mathcal{A}$ is a particular realization (e.g. a set of

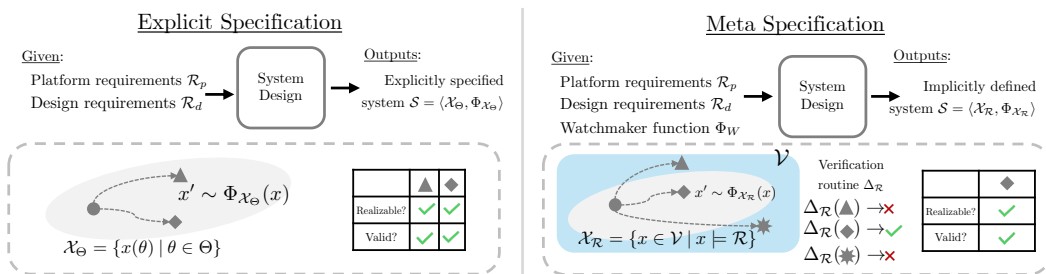

Figure 1: **Comparison of design principles.** Explicit specification (left) fully defines a system's state space and evolution function. Meta specification (right) implicitly defines the system's state space and evolution function by constraining a generalized representation space and employing watchmaker functions.

weights). More specifically, we make a distinction between the genotype space $\mathcal{G}_A$ and the phenotype space $\Pi_A$, which are related by a genotype-phenotype mapping $\pi : \mathcal{G}_A \to \Pi_A$ (Alberch, 1991). The genotype represents the agent's encoding that is evolved, while the phenotype represents the behaviors that emerge from its encoding. The subsystem's state space is its genotype space $\mathcal{A} \coloneqq \mathcal{G}_A$ as generally, evolution acts on the genotype, although our ultimate interest lies in novel phenotypic behavior. The *evolution function* $\Phi_A : \mathcal{A} \times \mathcal{T} \to P(\mathcal{A})$ is a stochastic process that evolves the agent based on interactions with the task. Here, $P(\mathcal{A})$ denotes the space of probability distributions over $\mathcal{A}$.

2. **Tasks.** Tasks are environments and objectives with which agents interact. The subsystem is similarly defined by a tuple $\mathcal{S}_T = \langle \mathcal{T}, \Phi_{\mathcal{T}} \rangle$. $\mathcal{T}$ is the state space encompassing all potential configurations of tasks that agents might encounter, where $t \in \mathcal{T}$ is a particular task. $\Phi_{\mathcal{T}} : \mathcal{T} \times \mathcal{A} \to P(\mathcal{T})$ is the task evolution function, which stochastically evolves new tasks based on the current task and agent states. Without loss of generality, each task can be considered as the combination of an environment and a goal. For example, it could be reaching a goal state in a Markov Decision Processes (MDP) (Bellman, 1958), or deriving the correct answer for a mathematical problem.

---

**Example 2.1: Underlying dynamical system**

In POET, **agents** are bipedal robots with fixed morphology and neural controller architectures. $\mathcal{A}$ and $\mathcal{G}_A$ is the weight space of the neural controller, which maps to locomotion behaviors (in the phenotype space $\Pi_A$). $\Phi_A$ is a stochastic weight update function (i.e. the evolution strategy algorithm (Hansen et al., 2015)) that evolves weights based on interaction with tasks. Additionally, **tasks** are MDPs with different terrains, controllable by $n$ free parameters that influence terrain shape. As such, $\mathcal{T} \subseteq \mathbb{R}^n$ is the space of tasks. $\Phi_{\mathcal{T}}$ evolves tasks by first selecting eligible tasks (i.e. have been solved), then introducing random mutations to the environment encoding.

---

The two subsystems are coupled, and often evolve *asynchronously*, where tasks can be evolved first, and agents are evolved subsequently by learning on the evolved task. The complete dynamical system is then formally defined as $\mathcal{S} = \langle \mathcal{X}, \Phi_{\mathcal{X}} \rangle$, where $\mathcal{X}$ is the Cartesian product of the agent and task spaces $\mathcal{X} = \mathcal{A} \times \mathcal{T}$ and its evolution function is the pair $\Phi_{\mathcal{X}} = (\Phi_A, \Phi_{\mathcal{T}})$. At this point, we recognize that, provided with the initial conditions, the dynamical system is a *fully defined* and simulatable. However, the direct evolution of such a system is not meaningfully interesting, as it lacks mechanisms for promoting open-ended progress. For example, agents could interact with randomly evolved tasks that are trivial, redundant, or overly difficult, leading to stagnation or degenerate behavior.

**Population-based evolution.** While we have, so far, focused on a single agent-task pair, OELS often involves populations evolving simultaneously. This does not alter the evolution of each pair, as all pairs could be considered as evolving in parallel. In what follows, we will describe the system as operating on a population of agent and tasks. In the interest of completeness, we will also mention that different pairing strategies could be used. Specifically, ▶ 1-to-1 pairing: each agent paired with one task (Wang et al., 2019); ▶ 1-to-M pairing: each agent paired with multiple tasks (Bauer et al., 2023); or ▶ M-to-1 pairing: multiple agents paired with one task (Mouret & Clune, 2015). The pairing strategy influences intra-system dynamics, for example, by promoting specialization,

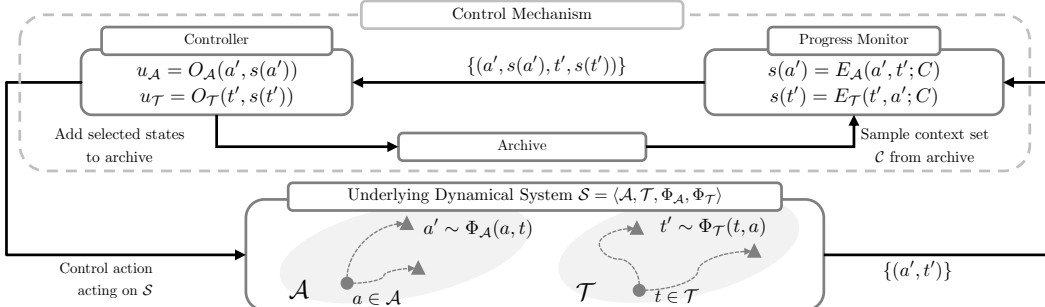

Figure 2: **Overview of OELS.** Conceptualization as a closed-loop system, where the control system monitors progress in the underlying system, taking control actions to guide continuous progress.

generalist behavior, or multi-agent competition. For simplicity, we assume 1-to-1 pairing, though our formalism generalizes to other settings with appropriate modifications to the evolution functions.

## 2.2  CONTROL MECHANISM

The goal of the control mechanism is to guide the system towards continuous open-ended progress. It constitutes two key components: a **progress monitor** that evaluates the agents and tasks produced by the dynamical system in each step to measure some *notion of progress*; and a **controller** that processes these measurements and takes control actions to enhnace continued progress.

**Notions of progress.** Various control mechanisms have been proposed based on different conceptions of what constitutes meaningful progress in an open-ended setting. In general, progress is measured relative to an aggregation of recent entities produced by the system (either historically or in the current population), which we refer to as the **context set**. Formally, we define the context set as $C \in \mathcal{C}$, where $\mathcal{C} = \mathbb{P}(\mathcal{A}) \times \mathbb{P}(\mathcal{T})$, and $\mathbb{P}(\cdot)$ denotes the power set and $\mathcal{C}$ is the Cartesian product of these power sets. More concretely, evolved agents or tasks are compared explicitly or implicitly (e.g. in an amortized fashion) against the context set to obtain a metric of progress. Examples of notions of progress include novelty (Lehman & Stanley, 2011; Stanley & Lehman, 2015), complexity (Standish, 2003; Hintze, 2019), learnability (Schmidhuber, 2013; Matiisen et al., 2019), diversity (Mouret & Clune, 2015; Pugh et al., 2016), or interestingness (Zhang et al., 2024). We provide a detailed review of different notions of progress in App. B.

1. **Progress monitor.** The monitor evaluates outputs of the dynamical system using these operational measures of progress, which we can explicitly define as two evaluation functions: $E_{\mathcal{A}} : \mathcal{A} \times \mathcal{T} \times \mathcal{C} \to \mathbb{R}$ (for agents) and $E_{\mathcal{T}} : \mathcal{T} \times \mathcal{A} \times \mathcal{C} \to \mathbb{R}$ (for tasks). Here, we have made explicit the monitor's evaluations are conditioned on a dynamically updated context set to capture metrics of non-stationary progress.

2. **Controller.** Controllers generally take as input the current set of agents and tasks and their corresponding scores produced by the progress monitor, differing primarily in the control actions they execute. A common strategy is *selection*, which directly selects the next set of inputs for the dynamical system to evolve (Brant & Stanley, 2017; Wang et al., 2019; Bauer et al., 2023), embodying the principle of *differential reproduction* observed in natural evolution and emulated in evolution-inspired algorithms (Gregory, 2009; Holland, 1992). More broadly, controllers may take any action that influences system dynamics, such as updating parameters of the underlying system's evolution functions (Wang et al., 2019; Zhang et al., 2024). Formally, we define two controller functions $O_{\mathcal{A}} : \mathbb{P}(\mathcal{A}) \times \mathbb{P}(\mathbb{R}) \to \mathcal{U}$ and $O_{\mathcal{T}} : \mathbb{P}(\mathcal{T}) \times \mathbb{P}(\mathbb{R}) \to \mathcal{U}$, which take as input the current set of agents (or tasks) and their corresponding evaluations, and produce a control action $u \in \mathcal{U}$ that acts on the dynamical system, where $\mathcal{U}$ represents the set of control actions.

> **Example 2.2: Control mechanism**
>
> In POET, the **progress monitor** evaluates agents and tasks using the current population as the context set. It first checks the *minimal criterion*, ensuring each agent solves at least one task and each task is solvable by one agent, then calculates a *novelty score* for qualifying pairs. The **controller** removes pairs that do not meet the criterion and selects those for evolution based on novelty scores. It also periodically transfers and adapts agents to different tasks.

## 2.3 Key Insights

This unified framework and conceptualization of OELS yields a few key insights. First, the control mechanism establishes a critical **feedback loop**, where it continuously monitors progress within the dynamical system, and uses these evaluations to make adjustments that support ongoing, open-ended progress. This enables the system to evolve adaptively, responding to new developments within the system (Soros et al., 2017; Jiang et al., 2023). Second, we can differentiate the roles of the dynamical system and the control mechanism. Notably, the dynamical system defines the **space of possibilities** (represented by $\mathcal{A}$ and $\mathcal{T}$) and governs how agents and tasks evolve within this space (through $\Phi_{\mathcal{A}}$ and $\Phi_{\mathcal{T}}$). In contrast, the control mechanism guides continuous progress within this predefined space, but it cannot alter the fundamental limits imposed by the dynamical system's design.

**A unifying framework.** We note that the formalism presented here is intended to be generalized, at the compromise of perfect specificity for any particular implementation. While certain OELS may have distinctive features that do not map exactly onto our framework, its generality facilitates broader discussions and comparisons of OELS. In App. A, we show that, despite their individual characteristics, many OELS can be effectively described and analyzed within this framework.

## 3 Design of the Underlying System

The presented framework allows us to separate the design of OELS into two categories: the design of the underlying dynamical system and the control mechanism. The designs have fundamentally different implications—the design of the underlying system defines the space of possibilities and mechanisms for evolution, whereas the design of the control mechanism is aimed at enhancing and sustaining open-ended progress. We also note that the two design of the two components can be discussed independently, as the design of the underlying system determines foundational constraints, while the control mechanism (and their operationalized notions of progress) are generally applicable.

Our work investigates the design of the **underlying dynamical system**, which defines the range of potential agent behaviors and task configurations. The goal of the system design process, then, is to create a sufficiently diverse and rich space of possibilities that can foster the emergence of novel behaviors or increasingly capable agents. Additionally, the designed system would have to be feasible, which, concretely, is defined as the satisfaction of design constraints. Formally, the system design process receives a set of requirements $\mathcal{R} = \mathcal{R}_p \cup \mathcal{R}_d$, where $\mathcal{R}_p$ represent the set of constraints required by the *platform* that the OELS is executed on, and $\mathcal{R}_d$ represent design constraints set by the system designer based on the goal of the OELS. The platform is most commonly a computing environment, but can be any substrate, including the physical world where agents and tasks interact.

The *goal* of system design is to engineer a sufficiently diverse space of possibilities to support open-ended learning. The design process outputs a system $\mathcal{S} = \langle \mathcal{X}, \Phi_{\mathcal{X}} \rangle$ that satisfies the following properties:
1. **Realizability:** Any possible state and its evolved states are realizable (implementable) on the underlying platform, $\forall x \in \mathcal{X}, \forall x' \in \mathrm{supp}(\Phi_{\mathcal{X}}(x)) : x \models \mathcal{R}_p \wedge x' \models \mathcal{R}_p$.
2. **Validity:** Any state and its evolved states are valid and satisfy the designer requirements, $\forall x \in \mathcal{X}, \forall x' \in \mathrm{supp}(\Phi_{\mathcal{X}}(x)) : x \models \mathcal{R}_d \wedge x' \models \mathcal{R}_d$.

### 3.1 Explicitly Specified System Design

The conventional approach to system design is based on **explicit specification**, formally defined as:

> **Explicitly Specified Systems**
>
> An explicitly specified dynamical system is one where its state space and evolution function are fully defined at *design time*, and kept fixed during its *run time*. Specifically, this involves:
>
> 1. Defining an appropriate state representation. Formally, $\mathcal{X}_{\Theta} = \{x(\theta) \mid \theta \in \Theta\}$ and $\theta$ is a representation that defines each state $x(\theta)$, and $\Theta$ is the representation space that determines the full state space $\mathcal{X}_{\Theta}$.
> 2. Specifying the input and output domains, and the functional form of the evolution function, which maps the state representation to a distribution over next states, $\Phi_{\mathcal{X}_{\Theta}} : \mathcal{X}_{\Theta} \to P(\mathcal{X}_{\Theta})$.

For example, the space of possible tasks could be parameterized by a real vector, $\Theta \subseteq \mathbb{R}^n$, that encodes all possible configurations (e.g. height and distribution of obstacles in an obstacle course). The evolution function is a fully defined mathematical function that operates over this representation (e.g. performing random mutation of the real vector). We note here, that expanding the space of possibilities corresponds is then achieved by engineering more degrees of freedom into the system.

The underlying system in all current OELS are designed using this principle (Wang et al., 2019; Dennis et al., 2020; Team et al., 2021), which has recently been operationalized to an impressive degree by Bauer et al. (2023), which introduced a significantly upscaled state space containing 25 billion unique tasks. A key advantage of this approach is that it often guarantees system realizability and validity *by design*. Since all system components are defined upfront, the system is built to meet platform and design constraints, ensuring it functions as intended without unforeseen runtime issues. Additionally, explicitly specifying the system offers designers a high level of *foresight* and control, allowing them to embed their knowledge or assumptions through design (e.g. specifying promising morphological spaces for robotic agents).

However, a significant challenge arises when attempting to *scale* such systems to explore a broader space of possibilities. This is especially critical as the design of the system directly constrains the emergence of novel behaviors and capabilities within certain bounds and structures. As the system grows, the complexity of the design process increases exponentially, requiring designers to optimally introduce and balance more degrees of freedom. Despite the extensive engineering effort invested in Bauer et al. (2023)'s system, progress eventually plateaued. To unlock further progress on a larger scale would require an even more complex design process, underscoring the inherent difficulty of scaling using this approach.

## 4 WATCHMAKER FUNCTIONS AND META SPECIFICATION

A key challenge facing OELS is how to significantly expand the space of possibilities permitted by the design of the underlying system. This work proposes an alternative design principle termed **meta specification**. At its core, meta specification defines a system implicitly through constraints that must be satisfied to yield valid states, rather than explicitly enumerating the state space.

More formally, we can contrast explicit and meta specification. Whereas an explicitly specified system completely describes all possible states as $\mathcal{X}_\Theta = \{x(\theta) \mid \theta \in \Theta\}$, meta specification implicitly defines the state space through constraints: $\mathcal{X}_\mathcal{R} = \{x \in \mathcal{V} \mid x \models \mathcal{R}\}$. Here $\mathcal{V}$ denotes the universal set, which conceptually refers to the set of all possible elements under consideration. For example, $\mathcal{V}$ could be the set of all possible robot morphologies or learning environments (which Clune (2019) defines as *Darwin Complete*). Performing meta specification then requires three key components: **(1)** a generalized representation space $\mathcal{V}$ for agents and tasks, **(2)** a mechanism to perform evolution over these generalized representations $\Phi : \mathcal{V} \to P(\mathcal{V})$, and **(3)** a method to verify constraint satisfaction, i.e. $x \models \mathcal{R} \, \forall \, x \in \mathcal{V}$.

Regarding **(1)**, sufficiently generalized representations do exist for many domains. For example, Unified Robot Description Format (URDF) can practically represent a wide array of robot morphologies (Quigley et al., 2015) and PyTorch can represent a vast space of neural network architectures (Paszke et al., 2019). Additionally, many requirements can be formally or empirically verified, such as kinematic feasibility for robot designs or certain properties for neural networks. Hence, the most challenging component of meta specification is the **generalized evolution function**. In explicitly specified designs, evolution functions are well-defined mathematical functions with clearly specified input and co-domains, which guarantees they are well-behaved over the entire state space. Here, the space that the generalized evolution function operates over is no longer explicitly defined, meaning that it loses any guarantee to be well-behaved. This lack of any guarantees on outputs produced by such functions necessitates verification routines to ensure that outputs are both valid and realizable.

As such, the implementation of meta specification is predicated on the existence of a generalized evolution function and verification routines for a chosen generalized representation space. We term this class of generalized evolution functions as **watchmaker functions**, inspired by the "watchmaker" analogy in evolution (Dawkins, 1986).

## 4.1 Watchmaker Functions

Watchmaker functions represent a class of functions that can take various forms depending on the chosen generalized representation $\mathcal{V}$, but must satisfy certain conditions:

> **Watchmaker Functions**
>
> For a given generalized representation sapce $\mathcal{V}$, a watchmaker function $\Phi_W : \mathcal{V} \to P(\mathcal{V})$ must satisfy the following *necessary* conditions:
>
> **(C1) Stochasticity:** $\forall\, v \in \mathcal{V}$, repeated applications of $\Phi_W(v)$ may yield different outputs.
>
> **(C2) Generalized transformation:** $\Phi_W$ is capable of producing meaningful transformations to any element $v \in \mathcal{V}$. Here, "meaningful" implies the function has an acceptable likelihood $\varepsilon$ of producing outputs $v' \in \mathcal{V}$ that are valid and realizable given requirements $\mathcal{R}$, i.e. $\mathbb{E}_{v' \sim \Phi_W(\cdot\,|\,v)}[p(v' \models \mathcal{R})] \geq \varepsilon$.

The conditions **(C1)** and **(C2)** serve distinct purposes in defining watchmaker functions. **(C1)**, the stochasticity condition, introduces variability and exploration into the system's evolution, preventing deterministic loops. **(C2)**, the generalized transformation condition, ensures that the function has an acceptable efficiency in producing valid and realizable outputs. This latter condition distinguishes viable watchmaker functions from purely stochastic processes which, while theoretically capable of generating valid and realizable states, are considerably inefficient (Borges, 1998; Eddington, 2019). It is worth noting that in the special case where $\mathcal{V} = \mathcal{X}$ is an explicitly defined representation space, a fully-defined evolution function can be considered a watchmaker function with perfect efficiency ($\varepsilon = 1$). A prime example of a potential watchmaker function is the human cognitive process. It can operate over generalized representations (e.g. natural language) and stochastically generate meaningful transformations to diverse stimuli. To make the connection to OELS more concrete, this could manifest as follows: a human might receive a robot morphology expressed in some domain-specific language (DSL), along with an instruction such as "improve joint mobility". With some acceptable probability, the human could then produce an output that satisfies the given requirements.

**FM watchmakers.** Importantly, another class of models that could potentially serve as watchmaker functions are foundation models (FM) such as *Large Language Models* (LLMs) (Brown, 2020; Chowdhery et al., 2023). These models, pretrained on vast amounts of data, function as efficiently stochastic generators capable of performing meaningful transformations across diverse domains. Their potential is evident in tasks such as code synthesis (Chen et al., 2021), program evolution (Ma et al., 2024; Lehman et al., 2023), and plan generation (Huang et al., 2022). As LLMs operate in the language space, they could serve as watchmaker functions for generalized representations that utilize language tokens, such as DSL like PyTorch for neural network architectures or URDF for robot descriptions. The implications of FMs as automated watchmaker functions are far-reaching, potentially enabling significant expansion of the space of possibilities in OELS, while reducing design complexity. In Sec. 5, we empirically investigate this feasibility, providing concrete evidence for the potential of LLMs as effective watchmaker functions in practice.

## 4.2 Implicit System Design Through Verification

Watchmaker functions, while powerful, lack inherent guarantees of producing well-behaved outputs. This limitation is particularly relevant for FM watchmaker functions in meta specification, where evolved outputs (e.g., robot morphologies) may not always be realizable or valid against design requirements. Recall that **(C2)** stipulates that, in expectation, there is a $\varepsilon$ probability that the watchmaker function produces valid and realizable outputs. Then, a viable approach to ensure requirements satisfaction is to introduce *verification* routines that operate on watchmaker functions outputs. Formally, given a set of $n$ requirements $\mathcal{R} = \{R_1, \ldots, R_n\}$, we define verification routines $\Delta_{\mathcal{R}} = \{\delta_1, \ldots, \delta_n\}$, where each $\delta_i : \mathcal{V} \to \{0, 1\}$ verifies if an output satisfies a particular requirement. Additionally, we say $\Delta_{\mathcal{R}}(x) = 1$ if $x \in \mathcal{V}$ passes all verification routines. This verification process, used in conjunction with the watchmaker function, *implicitly* defines a system:

Figure 3: **Emergent novelty.** Valid and realizable states evolved using LLM watchmaker functions.

---

**Implicit System Definition**

A watchmaker function $\Phi_W : \mathcal{V} \to P(\mathcal{V})$ and a verification routine $\Delta_\mathcal{R} : \mathcal{X} \to \{0, 1\}$, implicitly defines a system $\mathcal{S} = \langle \mathcal{X}_\mathcal{R}, \Phi_{\mathcal{X}_\mathcal{R}} \rangle$ as:

$$\mathcal{X}_\mathcal{R} = \{x \in \mathcal{V} \mid \Delta_\mathcal{R}(x) = 1\}, \qquad \Phi_{\mathcal{X}_\mathcal{R}}(x' \mid x) \propto \Phi_W(x' \mid x)\Delta_\mathcal{R}(x')$$

---

Intuitively, the system is implicitly defined through rejection sampling principles, to evolve states that are determined to be valid and realizable by the verification routines. This stands in contrast to explicitly specified systems, which embed constraints explicitly through system design, meta specification implicitly defines a system by enforcing constraint satisfaction a posteriori.

### 4.3 IMPLICATIONS IN FULL

**Design process.** Implementing a system through meta specification concretely entails three key steps: ▶ selecting a suitable generalized representation for agents and tasks; ▶ designing a FM-based watchmaker, for example through prompt designs or model finetuning, to improve their efficiency over the chosen representations; and ▶ developing verification routines to confirm satisfaction of system and design requirements. We note that meta specification is not likely to be universally applicable to all open-ended learning goals, and its viability depends on these three steps.

**Expanding the space of possibilities.** LLM-based watchmaker functions have the potential to significantly expand the space of possibilities, especially for certain representations. DSL representation is especially powerful, as the compositional nature of code allows for combinatorial expansion of possibilities, enabling emergence not explicitly pre-defined (Backus, 1978). Moreover, many DSL and programming language are Turing-complete representations, theoretically allowing the space of possibilities to encompass any computable function (Turing, 1936).

**Shifting design focus.** Meta specification shifts the focus of system design from explicit definition of system component from the ground up, towards developing robust verification routines to ensure realizability and validity. As such, it potentially presents a more tractable approach in domains where direct system engineering is prohibitively complex. This shift in design paradigm echos observations across various fields suggesting that verification can be less complex than generation (e.g. from NP-hard problems (Dantzig et al., 1954) to machine learning (Goodfellow et al., 2014) and software engineering (Dijkstra, 1976)). However, it's crucial to recognize that some requirements resist straightforward automatic verification, and ensuring robust, scalable verification in the face of emergent system developments presents a significant challenge.

## 5 AN ILLUSTRATIVE IMPLEMENTATION

In this section, we provide an illustrative implementation to support the viability of applying meta specification to designing systems for OELS. In this feasibility study, we focus on two key aspects

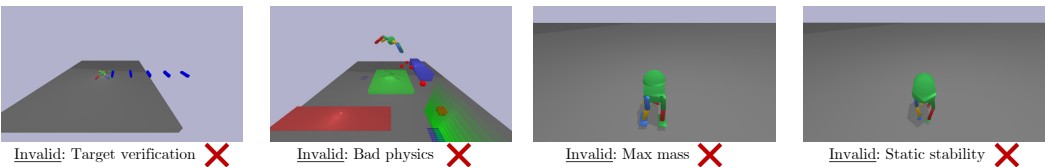

Figure 4: Examples of realizable but invalid tasks and robots.

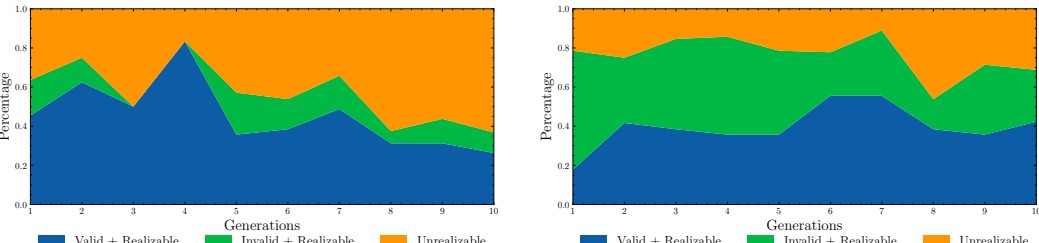

Figure 5: Proportion of task (left) and robot (right) evolutions that are valid and realizable.

1. **Emergent novelty** in robots and tasks generated through system evolution that were not explicitly preprogrammed, and
2. **Realizability and validity** of evolved outputs by way of the introduced verification routines.

We design a co-evolving system that simultaneously evolves both robots morphologies and robotic tasks. The spaces of robot morphologies and task configurations are especially difficult to encode in a rich, expansive way through explicit system specification, and as such are chosen to illustrate the value proposition of our proposed design approach. Concretely, we evolve a population of quadruped robots, where the robot's morphology is represented using URDF, an XML file format that defines a robot's physical design (Quigley et al., 2015). Simultaneously, we evolve a population of robotic tasks, represented as code programs in PyBullet, a physics engine commonly used for robotic tasks (Coumans & Bai). Thus, URDF and PyBullet provide the basis of our *universal representations*. We employ gpt4 as our LLM-based watchmaker function. Additionally, we introduce the following system constraints, designed to emulate the types of requirements typically imposed in real-world scenarios.

1. **Platform requirements.** Evolved morphologies must be valid URDF representations, and task configurations must be simulatable within PyBullet.
2. **Design requirements.** Robot morphologies are constrained to a quadrupedal configuration, with specific constraints on sensor types (restricted to proprioceptive sensors), the number and type of joints, mass, and size. Task designs are constrained with basic physics (e.g. gravity, friction, and restitution) and environment size.

These constraints are implemented as verification routines and also provided to LLM-based watchmaker functions as natural language instructions. We provide detailed descriptions of verification routines and prompt structures in App. C.

**Co-evolving system.** We employ a standard co-evolutionary algorithm in a minimalistic setting and avoid unnecessary design decisions that could obscure our investigation. The system co-evolves populations of robots and tasks, where the population at time step $n$ is represented as $\mathcal{P}_n = \{(a_n^{(j)}, t_n^{(j)})\}_{j=1}^{J_n}$, where $J_n$ is the size of the population at time step $n$. In this setup, each robot is paired 1-to-1 with a unique task. At each evolutionary step, a maximum of $N$ newly evolved robot-task pairs can be introduced into the population. The evolution of a new robot morphology is conditioned on $M \in \mathbb{N}$ parent pairs, where each parent consists of a robot and its corresponding task. Specifically, $a'_{n+1} \sim \Phi_W^{(i_a)}(\cdot \mid \mathcal{M}_n)$, where $\mathcal{M}$ denotes the set of parents sampled randomly from the current population, i.e. $\mathcal{M}_n = \{(a_n^{(m)}, t_n^{(m)}) \sim \texttt{Uniform}(\mathcal{P}_n) \mid \forall\, m \in [M]\}$. Here, $i_a$ refers to natural language instructions that contain morphology requirements. Similarly, tasks are evolved from $M$ parent task-robot pairs, i.e. $t'_{n+1} \sim \Phi_W^{(i_t)}(\cdot \mid \mathcal{M}_n)$, where $i_t$ encodes the requirements for the tasks. Robots and tasks evolved from the same set of parent pairs are then paired together. In each step, a set of candidate pairs are generated, where each pair is verified by verification routines, admitting only those that are realizable and valid into the population.

## 5.1 EMPIRICAL OBSERVATIONS

**Emergent novelty.** We visualize evolved robots and tasks in Fig. 3, observing a wide array of distinct and novel morphologies and tasks. The robots form niches ranging from ant-like creatures with long, slender legs to horse-like quadrupeds with sturdy limbs, as well as more unconventional forms resembling 'Walker' machines from Star Wars. We also observe interesting combinations of parent phenotypes, such as robots that retain ant-like legs but develop walker-like feet. The evolved tasks display similar diversity, including challenges focused on uneven terrain, obstacle courses, and constrained environments like mazes and tunnels. This emergent novelty occurs without any explicit preprogramming, demonstrating the system's capacity for generating complex and novel states.

**Role of verification.** In Fig. 4, we visualize several evolved states that were rejected for failing to meet the verification checks. This includes tasks where environmental objects violated physics constraints or where target locations were unreachable. For robots, examples of rejections included those that exceeded the maximum allowed mass or were unable to achieve static stability. In Fig. 5, we track the percentage of evolved outputs that are valid and realizable, noting that approximately 40% of the candidates evolved by the watchmaker functions satisfied both criteria. Interestingly, while the percentage of realizable robot morphologies was relatively high, the percentage of valid morphologies was lower. Based on our manual examination, we attribute this to several morphological constraints—such as static stability and mass—being phenotypic constraints rather than explicitly encoded in the URDF. As these attributes are only verified at runtime, it increases the likelihood that certain requirements will not be met, leading to a lower validity rate.

**Additional ingredients.** It is important to note that the implementation here is only the underlying dynamical system, which serves to illustrate the potential of an alternative system design principle. However, it would need to be complemented by additional components to fully realize it as an OELS. Most notably, we have omitted any training procedure (due to the cost and compute requirements), which is crucial for the evolved robots to become increasingly capable. Furthermore, a control mechanism should be integrated to ensure continuous progress. This could include regret-based approaches (Jiang et al., 2021) or LLM-based progress monitors (Zhang et al., 2024), or novel control mechanisms specifically tailored to the characteristics of watchmaker functions.

## 6 DISCUSSIONS

In summary, this work introduces meta specification as a novel approach to designing OELS. This principle enables implicit system definition through constraints, employing watchmaker functions in conjunction with verification routines to drive system evolution. In contrast to explicitly specified designs, meta specification offers the potential to significantly expand the space of possibilities while concurrently reducing system design complexity. Our illustrative demonstration of co-evolving robot morphologies and tasks illuminates the viability of this principle, showcasing its capacity for emergent novelty and underscoring the critical role of verification in maintaining system integrity. **Future directions.** Building on this foundation, subsequent research should prioritize scaling our proof-of-concept to large-scale implementations, thereby exploring the full potential of meta specification across diverse domains, including embodied agents and LLM-based reasoning systems. Furthermore, systems designed through meta specification could be augmented with complementary control mechanisms specifically tailored to foster the continuous generation of novelty and progress within this framework.

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

## A  INVESTIGATING OELS THROUGH A UNIFIED FRAMEWORK

The goal of this section is to illustrate that existing research in OELS can be conceptualized in the same unifying framework (presented in Sec. 2), as containing three key components—▶ **dynamical system** (co-)evolving agents and tasks, ▶ **progress monitor** that evaluates the evolutionary progress of current agents/tasks relative to a context set, and ▶ **control mechanism** that takes performs some control logic based on progress evaluations to sustain open-ended progress. We survey some of the most prominent and recent works, including Minimal Criterion Coevolution (Brant & Stanley, 2017), POET (Wang et al., 2019), PAIRED (Dennis et al., 2020), Ada (Team et al., 2021; Bauer et al., 2023), and OMNI (Zhang et al., 2024).

**Minimal Criterion Coevolution (MCC) (Brant & Stanley, 2017).** MCC co-evolves two populations: mazes and maze-solving agents, where individuals from both populations are evaluated for satisfying a minimal criterion (MC).

- **Agents.** Agents are maze-solving agents with evolvable neural controllers. The agent space $\mathcal{A}$ is the space of neural networks, varying in the number of neurons and connections. The agent evolution function $\Phi_A$ is the *NEAT* algorithm (Stanley & Miikkulainen, 2002), a genetic algorithm that evolves networks by adding connections or nodes.
- **Tasks.** Tasks are $2D$ mazes encoding using a variable length genotype. The task space $\mathcal{T}$ include mazes with varying number of walls, wall connections, and location of openings. The task evolution function $\Phi_T$ is a random mutation that modifies wall characteristics or number of walls.
- **Progress Monitor.** Progress is evaluated using satisfaction of minimal criterion, where the context set is the current population. Specifically, each maze-solver satisfies the MC if it solves at least one maze in the context set, and each maze satisfies the MC if solvable by at least one maze-solver.
- **Controller.** The control logic removes agents and tasks that do not satisfy the MC, with the remaining population fed into the system for another round of evolution.

MCC is similar in principle to 'differential reproduction with variation' observed in natural selection and emulated in evolutionary algorithms (Holland, 1992; Gregory, 2009). In other words, the control mechanism evaluates for fitness (using the MC) and performs selection, with fit individuals allowed to reproduce and generate variations in the system.

**POET (Wang et al., 2019).** POET co-evolves populations of bipedal robots and locomotion tasks in a $2D$ terrain.

- **Agents.** Agents are bipedal robots with identical morphology and neural controller architectures. The agent space $\mathcal{A}$ is the space of neural weights, which are evolved. The agent evolution function $\Phi_A$ is the *Evolution Strategies* algorithms, which evolves continuous vectors through genetic operations (Hansen et al., 2015).
- **Tasks.** Tasks are $2D$ terrains with different terrains and obstacles. The task space $\mathcal{T} \subseteq \mathbb{R}^4$ includes terrains with $4$ degrees of freedom (that control the size and frequency of obstacles). Task evolution $\Phi_T$ is performed through random mutations, where tasks eligible to produce are evolved to generate offspring tasks.
- **Progress Monitor** Progress is evaluated using MC and novelty, where the context set is the current population. Novelty of evolved tasks is calculated using the Euclidean distance of its environment encoding with its k-nearest neighbors.
- **Controller.** The controller selects offspring tasks with the highest novelty scores and admits them into the population. It also performs periodic attempts to transfer agents between different environments, to encourage cross-pollination.

**PAIRED (Dennis et al., 2020).** PAIRED trains a maze-solving agent in different environments. It differs from previous approaches by using an *environment-generating policy* to generate adversarial tasks to guide agent learning.

- **Agents.** Agents are maze-solving agents with fixed neural architectures. The agent space $\mathcal{A}$ is the space of possible neural weights. The agent evolution function $\Phi_A$ is an RL-based learning algorithm that evolves agent weights.
- **Tasks.** Tasks are mazes with different layouts in a *gridworld* platform. The space of tasks $\mathcal{T}$ is the space of maze layouts, where each tile could be a wall, start position, end position, or

pathway. The task evolution function $\Phi_T$ is the environment generating policy $\Lambda : \Pi \to \Delta(\mathcal{T})$, that produces a distribution over tasks given the current agent policy.

- **Progress Monitor.** The monitor evaluates the regret of the agent policy $\pi_a$ relative to a baseline policy $\pi_b$, i.e. $\text{REGRET}(\pi_a, \pi_b) = U(\pi_b) - U(\pi_a)$, where $U(\cdot)$ is the reward obtained by each policy. Here, progress is not evaluated with respect to an explicit context set; instead, it is *amortized* through $\Lambda$, which serves as a learned representation, implicitly encoding the capabilities of the current agent.
- **Controller.** The control mechanism updates the environment generating policy $\Lambda$ based on the regret of the agent and the baseline policy, training it to maximize regret, and generate more challenging tasks.

**Ada (Team et al., 2021).** Ada aimed to develop highly adaptable RL agents in an embodied $3D$ domain.

- **Agents.** Agents are embodied agents with fixed morphology and neural controllers. The agent space $\mathcal{A}$ is the space of possible neural weights. The agent evolution function $\Phi_A$ are meta-RL updates (Hessel et al., 2021).
- **Tasks.** Tasks are embodied and potentially multi-agent environments with procedurally generated goals. The task space $\mathcal{T}$ is one of the largest scale to date, containing $25B$ unique tasks, each procedurally generated by sampling from a parametric distribution over worlds, topologies, games, and opposing agents. The task evolution function $\Phi_T$ is based on random sampling, where a set of $J$ tasks are sampled randomly $\{t^{(j)} \mid t^{(j)} \sim P(\mathcal{T}) \, \forall \, j \in [J]\}$.
- **Progress Monitor.** The monitor assigns a fitness score to each randomly sampled task that approximates the agent's regret for that task, indirectly reflecting the learnability of the tasks. **Controller.** The controller selects tasks with fitness scores above a certain threshold.

**OMNI (Zhang et al., 2024).** Differing from prior works, OMNI employs an LLM as an evaluator of task novelty based on human notions of interestingness.

- **Agents.** Agents are RL agents with fixed neural controllers, and the agent space $\mathcal{A}$ is the space of possible weight configurations. Agent evolution ($\Phi_A$) occurs through RL updates.
- **Tasks.** OMNI investigated different task spaces, including $2D$ gridworld-like environments. The task space $\mathcal{T}$ is defined by parametric encoding of different tasks, where different task spaces are characterized by different free parameters. Task evolution ($\Phi_T$) occurs through a learning-progress-based curriculum (Kanitscheider et al., 2021).
- **Progress Monitor.** Progress is evaluated using an LLM's internalized notion of interestingness against a context set of recent tasks. The LLM predicts whether it finds evolved tasks interesting (i.e. a binary prediction).
- **Controller.** The control logic uses the LLM's evaluation of task interestingness to update task sampling weights in the curriculum.

## B   EXTENDED RELATED WORKS

**Notions of progress.** A key component in any OELS is the design of the control mechanism that is used to quantify and take control actions to foster various notions of what constitute *progress* in open-ended learning. Specifically:

- **Novelty:** Methods that encourage generations of agents or tasks that are sufficiently novel compared to previously seen examples (Stanley & Lehman, 2015). Novelty has been concretely formalized as Euclidean k-nearest neighbor (Lehman et al., 2008; Lehman & Stanley, 2011), or count of immediate neighbors in a discrete behavioral (*phenotypic*) space (Cully & Demiris, 2017). Such methods require a-priori for the behavior space to be manually defined and often discretized. Closely related to novelty-based approaches are those based on fostering diversity, which while conceptually distinct, are occasionally operationalized with similar metrics (Pugh et al., 2016; Mouret & Clune, 2015; Brant & Stanley, 2017).
- **Complexity:** Approaches that drive system evolution towards increasingly complex agents and challenging tasks (Standish, 2003; Hintze, 2019). For example, Kolmogorov complexity metric based on sliding-window compression of a sequence (Hintze, 2019) and JPEG compression of images to evaluate complexity (Earle et al., 2021).

- **Learnability:** Techniques that aim to balance task difficulty with agent capabilities to build an auto-curriculum for continual learning. Tasks are proposed within the agent's "zone of proximal development" to promote continuous growth and skill acquisition (Vygotsky, 1978). This approach is perhaps most common in the field of *Unsupervised Environment Design* (UED) which generates new RL training environments for agents (Dennis et al., 2020; Jiang et al., 2021; Parker-Holder et al., 2022; Samvelyan et al., 2023). These aim of such efforts is not true open-endedness, per se, are usually focused only on generating different variations in training environment to train robust agents, and a significant limitation is their reliance on predefined or manually curated distributions of tasks or environment parameters. Notable approaches use regret-based calculations (Dennis et al., 2020; Jiang et al., 2021; Parker-Holder et al., 2022) to prioritize tasks with high regret. Alternative methods calculate learning progress using learning curve slope (Matiisen et al., 2019) or differences in task success rates across training steps (Kanitscheider et al., 2021), or meta-learning potential (Team et al., 2021; Bauer et al., 2023).

**OELS.** An interesting array of different open-ended learning systems have been proposed, differing significantly in system design and intended application. *Chromaria* (Soros & Stanley, 2014) is a visual, 2D world composed of discrete RGB pixels, where colorful creatures (Chromarians) evolve and explore locations to plant, and is used to illustrate the necessary conditions for open-endedness to emerge in artificial life. More recent works have investigated systems aimed to fostering general capabilities through open-ended learning, including a 2D bipedal walker domain to improve robot locomotion (Wang et al., 2019), 2D grid worlds for hierarchical task completion (Dennis et al., 2020). Team et al. (2021); Bauer et al. (2023) introduced *XLand2*, containing 25-billion possible task variants corresponding to different world topologies and variety of possible games within each world. *Minecraft*, which contains procedurally generated 3D terrains and continuous exploration of technology trees (Wang et al., 2024). In App. A, we investigated these disparate implementations, demonstrating they could be analyzed according to our unifying framework.

**FM in OELS.** Recent investigations into FMs within OELS have primarily focused on their role in evaluating and fostering continued progress (i.e. as **control mechanism**). LLMs have been utilized as evaluators of qualitative notions of progress, assessing open-ended creativity of writing (Bradley et al., 2024) and interestingness of proposed states (Lu et al., 2024b; Zhang et al., 2024). While these approaches leverage FMs for evaluation and control, our proposal extends their application to the core system itself, with watchmaker functions potentially guided by these control components to sustain open-endedness. Although not developed in the context of open-ended learning, recent works have demonstrated the potential of LLMs as **generalized evolution** functions. This includes evolutionary search for code (Lehman et al., 2023; Ma et al., 2024; Chen et al., 2024) and in-context generation of meaningful variations (Meyerson et al., 2023; Fernando et al., 2024). Our approach builds upon these findings, proposing the potential of LLM watchmakers for meta specification of systems for open-ended learning. Parallel research has explored LLMs as **embodied agents** in open-ended settings, functioning as decision-makers and task executors for novel exploration (Wang et al., 2024; Lu et al., 2024a). While these studies focus on LLMs *as* agents within the system, our watchmaker concept extends to a broader context, encompassing systems that evolve diverse entities such as robot morphologies or algorithms, thereby offering a more generalizable framework for open-ended evolution. Additionally, research into training FM for generating action-controllable virtual worlds offer potential applications in OELS (Bruce et al., 2024; Earle et al., 2024).

## C  ADDITIONAL DETAILS ON ILLUSTRATIVE IMPLEMENTATION

In this section, we provide additional details on the verification routines and LLM-based watchmaker functions employed in our illustrative implementation in Sec. 5.

### C.1  VERIFICATION ROUTINES

**Robot constraints.** Evolved robots are represented as URDF files, which must describe quadruped robots that satisfy the following constraints:

1. **Realizability:** Realizability is checked by 'compiling' the URDF file in PyBullet, where verification fails if any compilation or runtime errors are encountered (indicating the file is malformed).

2. **Size and mass:** Length $\in [0.5, 2]$ meters, width $\in [0.5, 2]$ meters, and height $\in [0.25, 1]$ meters. Mass $\in [50, 250]$ Kgs.

3. **Joints:** Number of joints $= 8$, and all joints are revolute joints.

4. **Static stability:** The evolved robot is required to achieve static stability when no actions are applied. This is verified by deploying the robot in the seed task environment for $50$ settle steps, during which time, there should be no movements in the main torso.

**Task constraints.** Evolved tasks are represented as Python programs written using PyBullet, and satisfy the following constraints:

1. **Realizability:** Realizability is checked by 'compiling' the task code, where verification fails if any compilation or runtime errors are encountered (indicating syntax errors).

2. **Basic physics:** That gravity is correctly set to $9.8m/s$, and friction of any ground planes $\geq 0.8$ and restitution of any obstacles $\geq 0.5$.

3. **Initialization:** That upon initialization, the robot is successfully positioned at the intended starting position, where it can achieve static stability.

4. **Target verification:** For tasks with target locations, we verify that the target location exists by raycasting and confirming that ground planes extend to that location.

## C.2 LLM PROMPT DESIGN

In our illustration, we utilize OpenAI's `gpt4` LLM as watchmaker functions to perform agent and task co-evolution. The prompt skeleton for each operation is provided below.

```
You are an expert in Python programming and robot design, specializing
    in creating quadruped robots that can master diverse tasks in
    PyBullet simulations. Your goal is to design the next iteration of a
    robot, focusing on capability, novelty, and interesting features
    while adhering to specific constraints. You will be provided with
    the current robot morphology and the recently accomplished task code
    exmamples to help you design the next robot.

Instructions:
- Physical realism:
    - Ensure the design is implementable in PyBullet and is physically
        realistic.
    - The robot must be capable of completing various tasks.
    - The robot must have optimal stability and will not fall over.
- Novelty and creativity:
    - Introduce unique and innovative features compared to the current
        morphology.
    - Design should enhance the robot's capabilities for diverse tasks.

Constraints:
- The robot must have a base and four articulated legs.
- It must have exactly 4 legs, and each leg must have 2 hinge joints.
- The robot must achieve static stability, meaning the robot should be
    able to stand without falling over.
- The robot can only have proprioceptive sensors but no perception
    sensors. It can sense its own joint angles and joint velocities, but
    it cannot sense the environment or objects around it.
- The robot can have a total length and width between 0.5 - 2 m and a
    total height between 0.5 - 1 m.
- The robot can have mass between 50 - 200 kg.
- Everything else is up to you, be creative with the morphology of the
    robot.

Desired format:
Reasoning for what the next robot morphology should be:
<reasoning>

Next robot morphology:
```

```XML
<XML code>
```

Current task code:
{TASK_CODE}

Current robot XML:
{ROBOT_XML}

Listing 1: **Prompt for robot evolution.**

```
You are an expert in Python programming and PyBullet environment design.
    Your goal is to code an environment in PyBullet that a robot can
    train on to become generally capable. You will be provided with
    pairs of environment code and robot XML descriptions.

Instructions:
- Introduce environments that are novel, but not too difficult given the
    current environment the robot is trained on.
- The task should be learnable with 2 hours of RL training
- The environment must be implemented using PyBullet, do not use any
    other packages.

Constraints:
- The environment must use class name 'Env'.
- The environment must be suitable for the robot's physical size and
    capabilities. For example, any object that needs to be traversed
    should be at least twice the width of the robot for it to move
    around.
- Any target location or object should be within 10 meters of the
    robot's initial position.
- Given the target location, the robot should physically be able to
    reach it within the environment.
- The robot should be initialized with an orientation that aligns it to
    face toward the positive x-axis.
- Any randomly generated objects should be seeded to ensure
    reproducibility across different runs.
- The lateral friction of any object or terrain traversed by the robot
    should be set to 0.8, and the restitution should be set to 0.5.
- If you need to access PyBullet functions, use 'self._p' to call them,
    do not add additional search paths.
- The robot has proprioceptive sensors but no perception sensors. It can
    sense its own joint angles and joint velocities, but it cannot sense
    the environment or objects around it. Do not implement tasks that
    require the robot to perceive or see the environment.
- Use creative colours, textures, and shapes for objects to make the
    environment visually appealing.
- Always call 'self.create_visual_target_marker()' providing the target
    location for the task at the end of '__init__()'.

Desired format:
Environment code:
```python
<environment code>
```

Current robot XML:
{ROBOT_XML}

Current task code:
{TASK_CODE}
```

Listing 2: **Prompt for task evolution.**

