# OpenReview forum: "Watchmaker Functions and Meta Specification of Open-Ended Learning Systems"
_ICLR.cc/2025/Conference — ICLR 2025 Conference Withdrawn Submission_

### Official Review · Reviewer_NFS9 · 2024-10-28

**Soundness:** 3
**Presentation:** 4
**Contribution:** 1
**Rating:** 3
**Confidence:** 3

**Summary:**

The authors propose a general framework for open-ended (co-)evolution. They first formalize existing work in that domain under the term "open-ended learning systems". Then they propose "watchmaker" functions to enable evolution to operate over arbitrarily flexible representations (including language), rather than being constrained to what they call "explicit specifications".

A "watchmaker function" seems to designate any mutation operator with an acceptable chance of producing viable output, together with validation procedures to ensure that the mutated outputs are viable under the requirement of the experiment.


They then describe early experiments to illustrate these ideas in a 3D physical simulation, co-evolving both environments (tasks) and agents, based on language descriptions and large language models.

**Strengths:**

Open-endedness is an important subject that deserves more attention.

The experiments, although minimal, seem to point towards some novelty (I am not aware of any experiment co-evolving both tasks and agents, in a 3D world, with language descriptions and LLM-based evolution operators) and hint at potentially interesting future work.

The paper is very well written, both in terms of content and of the very pretty presentation.

**Weaknesses:**

**Main contribution**

As I understand it, the first 8 pages of the paper can be summarized as follows:

"If we want to explore spaces defined by more flexible representations than fixed-shape tensors, such as language, we need to add some checks to ensure the resulting proposals are feasible".

I don't disagree! Does it require a whole paper, with considerable novel notation and terminology?

To illustrate the problem, in line 302 we read:

> Whereas an explicitly specified system completely describes all possible states as XΘ = {x(θ) | θ ∈ Θ}, meta specification implicitly
defines the state space through constraints: XR = {x ∈ V | x |= R}. Here V denotes the universal
set, which conceptually refers to the set of all possible elements under consideration.

If the system is to be evolvable at all, it *must* be represented through some kind of parametrization. As such, in reality, both of these sets imply a "theta". The former one seems to imply that the explicitly defined set Cap_Theta contains only, and all, valid specifications, whereas in the latter we need additional checks to ensure validity. But almost all non-trivial open-endedness experiments are already of the second kind, and definitely involve such checks!

For example, systems based on Karl Sims' virtual creatures (1994!) can represent any non-cyclic morphology, including infinitely many unfeasible ones, and as a result, require filtering to reject unfeasible agents (e.g. self-intersecting). It seems superfluous to call tree operations followed by sanity checks "watchmaker functions". More recently Lehman et al. (arxiv 2206.08896) proposed to use LLMs to mutate and evolve simple robots represented in code, which of course required some a posteriori filtering.

Thus, the contribution of the paper seems unclear.

**OELS framework**

The authors also attempt to provide a formal specification of "open-ended learning systems", but this formalization seems confusing to me. It is not obvious what should be regarded as part of the "evolution functions", and what should be part of the "control system".

This is evident in the authors' own chosen illustrative example, namely Wang et al.'s POET. They define the "Control mechanism" as consisting of the minimum criterion and the novelty check. But these are just how new environments are created and selected, i.e. the environment "evolution function" ! (In fact the original POET paper defines this as "Mutate_envs", algorithm 3 in the appendix)

**Experiments**

The experiments bear strong resemblance to those of Faldor et al. 2024 (OMNI-EPIC: https://arxiv.org/abs/2405.15568 ), which also use LLMs and PyBullet. The main addition IIUC is that now the agent co-evolves with the task (this would potentially be a valid contribution if the experiments were the central focus of the paper, which doesn't seem to be the case here).

Yet Faldor et al. is apparently not cited in this submission, which seems to be a serious oversight.

[UPDATE] I see that Reviewer sx3L has also noted the strong similarity and the lack of citation. Even if this submission is from the same team, Faldor et al. 2024 must be cited!

**Minor:**

In line 334, shouldn't "generalized transformation" read "viability" or some such single noun to match "Stochasticity" ? (authors use the term 'viable' in line 343)

l331: sapce -> space

**Questions:**

Why do we need additional formalism, notation and terminology to describe what has been done for decades, namely, evolution over representations sufficiently flexible to produce non-viable outputs?

What is the exact difference between the "evolution function" and the "control mechanism", and can you update your description of POET to match the original paper?

Could you please cite Faldor et al. 2024 and briefly specify the difference?

---

### Official Review · Reviewer_sx3L · 2024-11-02

**Soundness:** 1
**Presentation:** 1
**Contribution:** 1
**Rating:** 1
**Confidence:** 5

**Summary:**

This paper introduces "meta specification", a novel approach to designing open-ended learning systems. While current systems rely on explicitly defining all spaces and functions at design time, which becomes prohibitively complex at scale, meta specification defines systems implicitly through constraints using "watchmaker functions" - generalized evolution functions coupled with verification routines. The authors propose Large Language Models as potential watchmaker functions and demonstrate their approach through a system co-evolving robot morphologies and tasks.

**Strengths:**

The strength of the paper is that it offers a practical solution to a significant scaling problem in open-ended learning systems.

**Weaknesses:**

- Plagiarism of existing ideas, see Details Of Ethics Concerns section
- Lack of novelty
- No empirical results

**Questions:**

No question.

**Details Of Ethics Concerns:**

Upon reading this paper, I noticed striking similarities to another paper, titled [OMNI-EPIC: Open-endedness via Models of human Notions of Interestingness with Environments Programmed in Code](https://arxiv.org/abs/2405.15568), available publicly on arXiv since 24 May 2024. The code of the paper was open-sourced on 30 August 2024 on [GitHub](https://github.com/maxencefaldor/omni-epic).

The similarities are significant enough to raise concerns about plagiarism, claiming another team’s ideas as one’s own and original, and at a minimum failure to properly attribute ideas and the use of another project’s code.

I believe that the paper not only heavily draws inspiration from OMNI-EPIC's ideas and overall narrative, but also appears to utilize OMNI-EPIC's publicly available codebase **without attribution**. The use of that codebase also proves that the authors could not have independently come up with these ideas, but instead were clearly aware of OMNI-EPIC's paper before embarking on their research project and write-up.

I have compiled a list of evidence for you to review: https://docs.google.com/document/d/e/2PACX-1vT0YI0nLOwhmIOVThfjR_Fxb1gvmq8H33n_KHIXa1iOoIg8UQREpfQfX3_9dS-rYVKmR05zfnuDQz9_/pub

---

> ### Author Response · Authors · 2024-11-17
> **Response to Reviewer sx3L**
>
> First of all, we sincerely thank the reviewer for raising these important concerns. In focusing on the novel aspects of our work, we unintentionally and inexcusably failed to acknowledge OMNI-EPIC and our use of their codebase. We explain below how this serious oversight occurred but we want to let you know upfront that we are withdrawing our paper immediately.
>
> **Citation oversight:** While our work cited OMNI (produced by the same group) several times as a foundational work, we failed to cite OMNI-EPIC directly because we incorrectly assumed that the OMNI citation covered it. Your review has made us see what we should have realized ourselves: that OMNI-EPIC is a significant independent contribution that deserved proper citation and discussion, not, as we wrongfully viewed, as extending the concepts presented in Section 5 of OMNI.
>
> **Different research focus:** Our paper's primary aim is theoretical in nature, focusing on (1) analyzing and formalizing system design principles of open-ended learning systems (Section 2); (2) formally analyzing and discussing limitations in current system design principles (Section 3); and (3) proposing meta-specification as an alternative design principle (Section 4). The last contribution introduced three key ingredients: generalized agent/task representation spaces, evolution mechanisms (watchmaker functions), and constraint verification for implicit search space definition. In our focus on developing these contributions, we failed to properly consider the importance of comparing against significant methodological innovations in OMNI and OMNI-EPIC. We should have referenced OMNI-EPIC and contextualized how our framework relates to both papers. This was a major failure on our part.
>
> **Implementation:** Our illustrative implementation in Section 5 built upon OMNI-EPIC's codebase (specifically the base environment class definitions) without proper attribution - this was an unacceptable oversight. Although our implementation includes several novel components (co-evolution of robot morphologies and tasks, co-evolutionary algorithms, verification/validation mechanisms, and independently designed prompts), we should have clearly attributed the foundational code elements to OMNI-EPIC's authors. Furthermore, although we attempted to do this, we should have better emphasized that our implementation served primarily to demonstrate our framework rather than advance methodological contribution.
>
> Although our errors were entirely unintentional, we regret not acknowledging that our work was built upon the work of others and that we did not give them proper accreditation. Once again, we apologize sincerely. As embarrassed as we are, we are grateful the conference's review process identified these oversights before publication. We are taking concrete steps to reflect on and revise our internal processes, to ensure such issues will not occur in the future.
>
> *Finally, we thank all reviewers and the conference organizers for their thorough review process and for maintaining the high standards expected in our academic community.*
>
> The Authors of #13036

---

### Official Review · Reviewer_MNYu · 2024-11-04

**Soundness:** 3
**Presentation:** 3
**Contribution:** 3
**Rating:** 5
**Confidence:** 4

**Summary:**

The paper introduces a novel framework for the design of Open-Ended Learning Systems (OELS), proposing an alternative design principle termed meta specification. Instead of relying on explicit specification, where state spaces and evolution functions are predefined, this approach uses constraints to define systems implicitly. Central to the framework is the concept of watchmaker functions—stochastic evolution functions coupled with verification routines to foster system evolution within generalized representation spaces. The authors showcase an implementation that co-evolves robotic morphologies and tasks, leveraging a LLM as a watchmaker function.

**Strengths:**

The meta-specification approach presents a unique perspective on OELS, shifting from an explicit to an implicit design, which could simplify design complexity while expanding the space of possibilities.

Additionally, the paper formalizes a unified framework for OELS, creating a common language that facilitates the comparison of different OELS approaches.

The implementation of co-evolving robotic agents and tasks showcases the potential of meta specification and watchmaker functions to create emergent behaviors without requiring highly specific design constraints.

The use of LLM-based watchmaker functions offer a promising approach for larger-scale OELS implementations.

**Weaknesses:**

I think the paper is interesting and it's a valuable addition to the open ended learning literature. However, I think the contributions of the paper could be made clearer. For example, compared to POET, can the proposed system solves anything that POET can not or is it mostly just a generalized formulation of a class of algorithms.

Additionally, for a paper on open-endedness, the paper does not really show any open ended learning. The authors note that "Most notably, we have omitted any training procedure (due to the cost and compute requirements), which is crucial for the evolved robots to become increasingly capable. Furthermore, a control mechanism should be integrated to ensure continuous progress." but I think showing an actual more open-ended learning process would have made the paper more impactful.

Minor comments:
- Typo: “representation sapce” in the watchmaker function definition

**Questions:**

Is it always possible and feasible to run the verification process? How difficult is it to design this part in comparison to traditional explicit specification?

Figure 5 only show very low generation numbers. Why was the system not run for longer?

---

### Official Review · Reviewer_UCYe · 2024-11-08

**Soundness:** 2
**Presentation:** 1
**Contribution:** 2
**Rating:** 1
**Confidence:** 2

**Summary:**

The author’s propose an alternative approach to tackling design problems to mitigate the limitations of defining explicit specifications (e.g. a reward signal). Instead, the authors argue for a framework which specifies meta constraints to evaluate designs against to verify their feasibility in practice. They describe an extensive abstract framework and propose the concept of “watch maker functions” as a means of handling meta specifications. They then show an example scenario using a larger language model as a watchmaker function for generating robotic designs. This work is motivated in advancing the development of open-ended learning systems that generate novel challenges to push the limits of learning agent systems.

**Strengths:**

Overall the manuscript reads quite coherently. The author’s proposition of defining constraints as opposed to target specifications is an interesting approach for addressing design problems. By having fewer specifications, it seems plausible the authors framework could offer nove alternatives for addressing a number of hard problems.

**Weaknesses:**

Currently, the paper needs to provide meaningful evidence of the utility of the meta-specification framework, whether mathematically or practically. There's a lengthy description of the problem formulation and a watchmaker's function. It is unclear what these definitions give us theoretically because the authors do not use the results to prove substantial results. The paper would benefit from deeper theoretical analysis beyond establishing a mathematical framework description.

In addition, the author's choice to exclude practical, more easily deployable examples of watchmaker functions is quite limiting. The current examples in the paper are humans or otherwise large language models. Given that the latter is a recent phenomenon, this idea of watchmaker functions as presented would mean that five years ago, the notion of a watchmaker function was constrained solely to human-level intelligence by the author's examples. With this in mind, we strongly encourage the authors to provide more accessible examples of applications of this definition in future versions of this work.

Furthermore, we recommend that the authors re-evaluate how they introduce large language models in this paper. In its current form, the introduction should include the concept more smoothly. It reads as being forced in as an afterthought despite seemingly being the only other way to deploy watchmaker functions. As the author's practical results use large language models as the central component in the existing illustrative example, large language models deserve better attention in the paper discussion.

In addition, the paper should have a proper experimental evaluation to justify this framework.  The current illustrative implementation section is not compelling, particularly given that several works on robotic agent design already exist, which don't require access to a foundation model. Perhaps this approach does have merits, but without more concrete evidence, we do not believe this paper helps support the author's arguments.

**Questions:**

- What are the potential theoretical benefits of having a watchmaker function?
- What are other more tangible examples of watchmaker functions?
- Are the costs the authors mean when they say they “omitted any training procedure”? If this is directly related to having used gpt-4, this seems a notable limitation of the idea of watchmaker functions if you must rely on larger language models to have a meaningful function for this purpose.

---

### Note · Authors · 2024-11-17

I have read and agree with the venue's withdrawal policy on behalf of myself and my co-authors.